# "Harms" Associated with Breast Cancer Screening and Reliability of Frozen Section in Older Women: In the Case of an 80 Year Old Woman

**Evangelia Antoniou** [1,*], **Stefanos Zervoudis** [1,2], **Andriani Vouxinou** [1], **Anastasia Bothou** [1,3], **Anisa Markja** [1], **Eirini Orovou** [1], **Panagiotis Tsikouras** [3] and **Georgios Iatrakis** [1]

[1] Department of Midwifery Athens, University of West Attica, 12243 Egaleo, Greece; szervoud@otenet.gr (S.Z.); avouxinou@univa.gr (A.V.); natashabothou@windowslive.com (A.B.); anisamrk@gmail.com (A.M.); eorovou@uniwa.gr (E.O.); giatrakis@uniwa.gr (G.I.)
[2] REA Hospital, 17564 Athens, Greece
[3] Department of Obstetrics and Gynecology, Democritus University of Thrace, 68133 Thrace, Greece; ptsikour@med.duth.gr
* Correspondence: lilanton@uniwa.gr; Tel.: +30-6972164739

**Abstract:** The objective of this paper is to present a rare case with negative final histologic examination despite abnormal findings of all previous exams indicating breast cancer in an 80 year old woman. Mammographic and magnetic resonance imaging findings were concordant with the frozen section biopsy result of DCIS. However, the final histologic diagnosis was radial scar (benign breast lesion that can radiologically mimic malignancy). As a conclusion, abnormal mammographic and magnetic resonance imaging findings with positive for DCIS frozen section reports are not always confirmed in the final histologic examination. Furthermore, considering that screening does not seem to be associated with a reduction in mortality due to breast cancer after the age of 75, breast cancer screening could be individualized in this age group.

**Keywords:** breast cancer screening; mammography; MRI; frozen section; DCIS

## 1. Introduction

Screening mammography has dramatically increased the detection rate of Ductal Carcinoma In Situ [1,2]. Risk factors for Ductal Carcinoma In Situ (DCIS) are similar to those for invasive cancer, including weight and age [3–7] among many others [8]. However, the potential benefits of screening mammography among women aged ≥75 years old remain unclear [9] and other imaging methods could be more suitable in certain cases [10]. Furthermore, considering that after a decade of annual screening, about 50 percent of women experience a false positive result [11], increasing the years of screening could increase the possibility of false positive results (although age is negatively correlated to breast density).

There are no specific clinical manifestations for patients with DCIS and most patients with a mammogram suggestive of DCIS mostly have no breast-related findings on physical examination. In most cases, DCIS corresponds to BI-RADS 4C category in mammography. However, a lesion of BI-RADS 4B or BI-RADS 4A could be the initial finding [12]. DCIS treatment includes mastectomy or breast-conserving therapy followed by adjuvant radiation in most cases. In cases with abnormal mammographic findings, with no core biopsies taken before the final surgery [13], extemporaneous examination/frozen section (FS) during the surgical procedure is extremely important, in order to rapidly provide the surgeon with a diagnosis likely to modify the course of the surgical procedure.

## 2. Case Presentation Section

Here, we present an 80 year old patient (with a body mass index of 24.6 kg/m$^{2)}$) who is included in the material of the third author's thesis. The screening mammogram, diagnostic MRI (to clarify mammogram findings) and specimen mammogram (Figure 1, Figure 2A–C and Figure 3, respectively) revealed an abnormal finding in the outer quadrant of the left breast. Architectural distortion (as the main finding), and some granular calcifications with irregularity in density, shape and size gave a BI-RADS 4C in the final imaging conclusions. During the surgery, the specimen sent to the pathology department revealed a DCIS initial diagnosis. However, in the final histologic report, no malignancy was found, and the definitive diagnosis was radial scar (a benign breast lesion that can radiologically mimic malignancy).

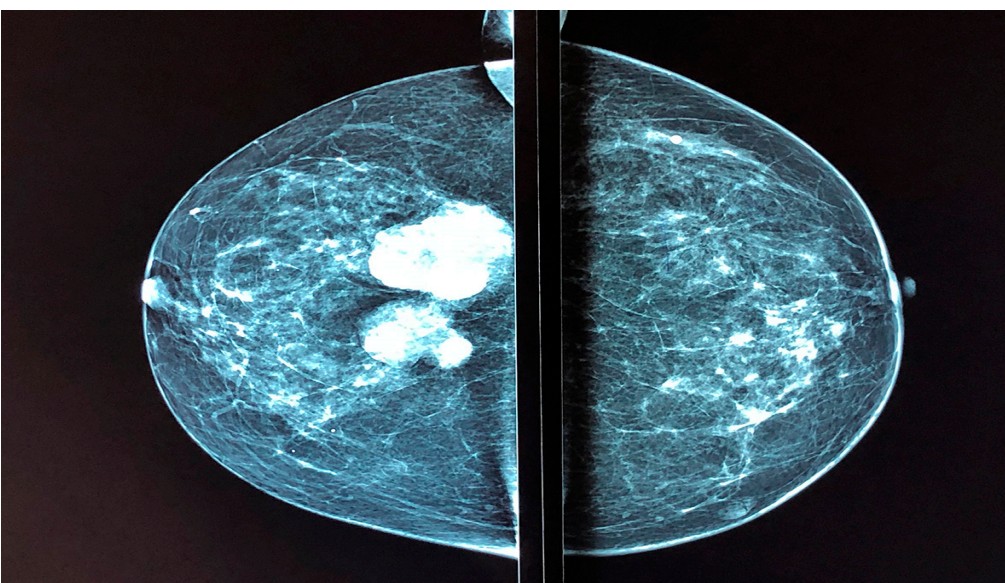

**Figure 1.** Screening mammogram.

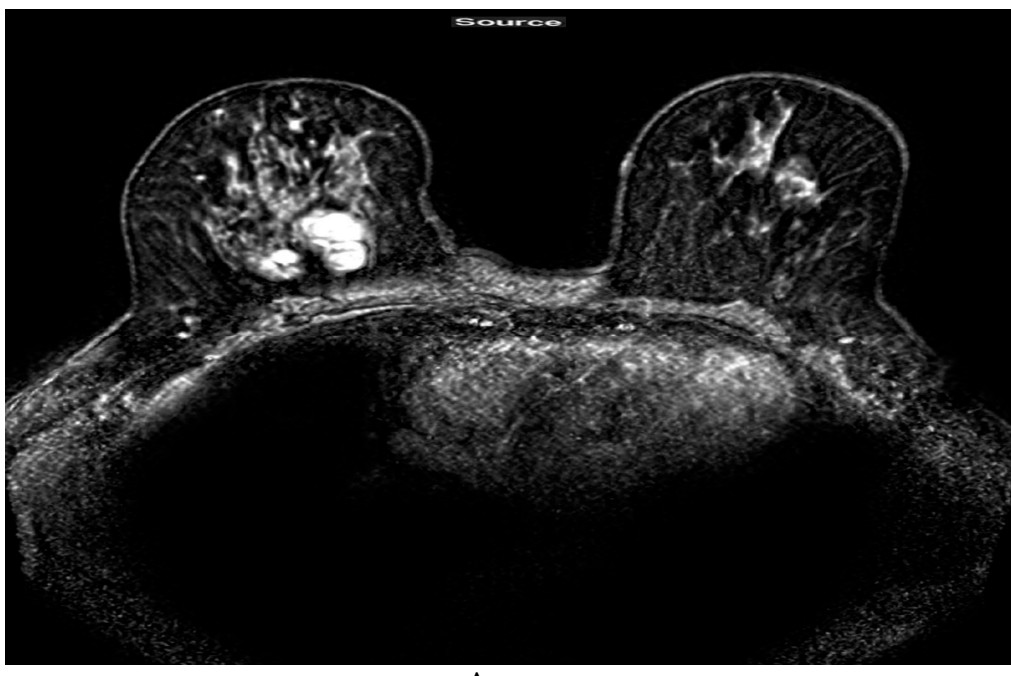

**A**

**Figure 2.** *Cont.*

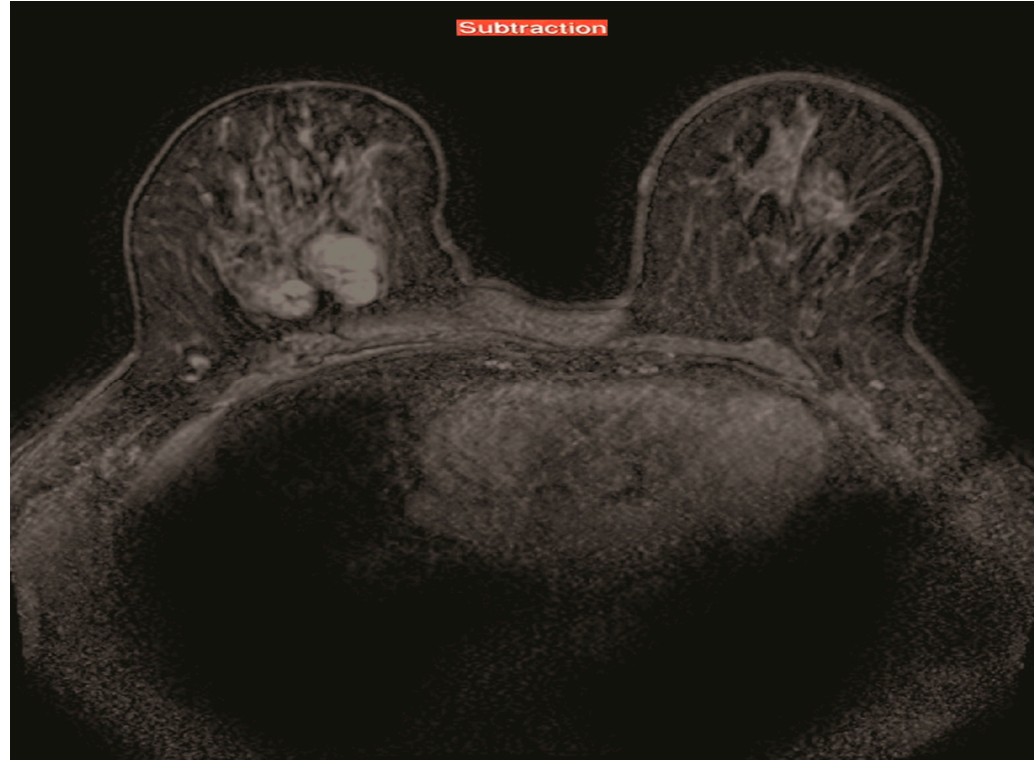

**B**

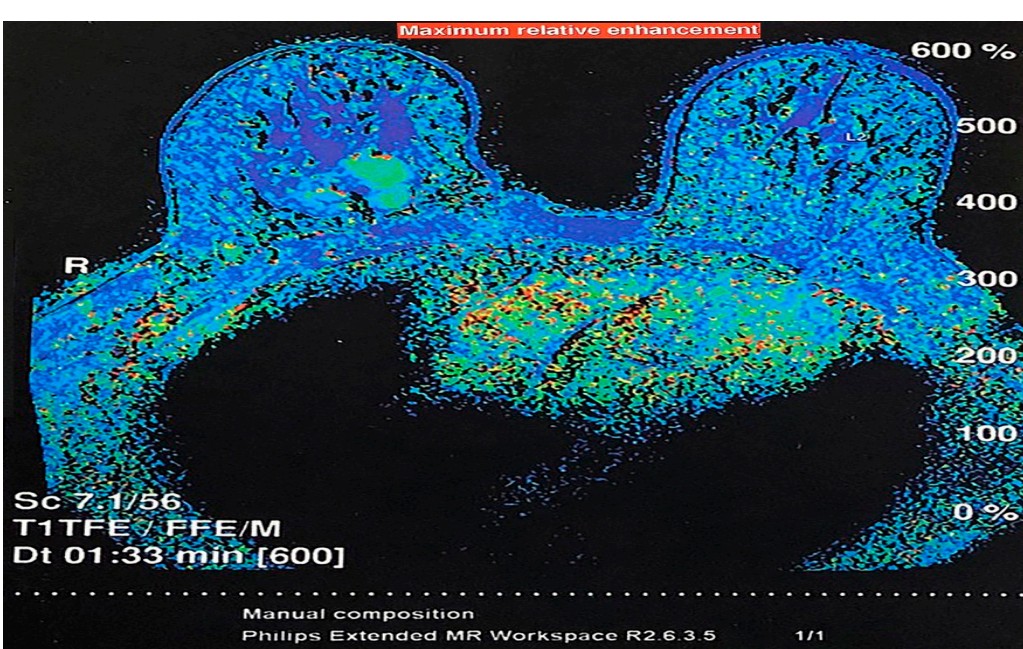

**C**

**Figure 2.** (**A**) Diagnostic MRI (Source). (**B**) Diagnostic MRI (Subtraction). (**C**) Diagnostic MRI (Maximum relative enhancement).

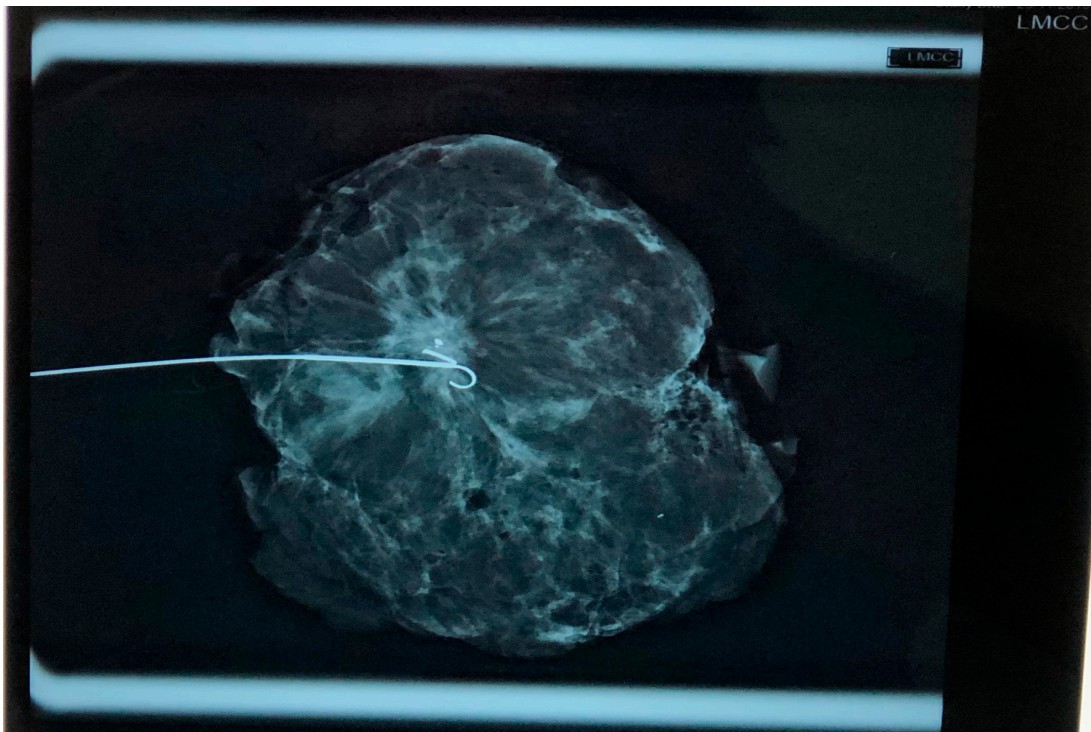

**Figure 3.** Specimen mammogram.

*Ethics Approval, Statement of Compliance and Clinical Trial Transparency*

The patient in this report provided consent for participation in this study which was approved by the Greek-French Breast Unit of Rea Hospital (Athens, Greece). The patient in this report provided consent for the anonymous publication of the data. All authors of this report agree with and are greatly obliged to the Editorial Board for the publication of this report.

## 3. Discussion

In breast pathology, FS is very sensitive, ensuring the diagnosis of malignancy without false positive results [14,15] in almost all cases. In our case report, both screening mammogram and diagnostic MRI (to clarify mammogram findings) revealed an abnormal finding of the breast, giving a BI-RADS 4C in the final conclusion. However, based on the lack of randomized clinical trials showing a benefit of presurgical breast MRI in overall survival, its integration into breast surgical operations remained debatable [16]. Furthermore, preoperative MRI for staging women with breast cancer does not reduce the risk of local or distant recurrences [17]. It could be claimed that the use of preoperative MRI is controversial. Actually, while prior studies indicated limited benefit of preoperative MRI, emerging evidence demonstrates lower odds of positive resection margins and lower odds of repeat surgery in patient groups undergoing preoperative breast MRI. In particular, from a clinical trial (involving authors from Yale School of Public Health and the Department of Surgery, University of Minnesota Medical School) involving more than 45,000 women having early stage breast cancer (with more than 9000 DCIS), it was concluded that in older women, preoperative breast MRI does not reduce the incidence of multiple surgeries [18]. However, from a recent study (involving authors from the University of Ulsan College of Medicine, Gangneung, Korea) involving less than 550 women, it was concluded that preoperative MRI reduced positive resection margins and repeat surgery for DCIS [19]. Considering that the authors of the latter study are deriving exclusively from two Departments of Radiology, a possible bias cannot be excluded.

In our case, frozen section confirmed the initial mammogram and MRI abnormal findings giving a DCIS diagnosis. However, the final histologic examination found no malignancy in the same area

of the breast. Although in this particular patient, the FS did not alter the next steps of the surgical procedure, a false positive FS result (in particular with an invasive cancer diagnosis) could dramatically affect surgical decisions. As an example, a DCIS diagnosis may entail "radical" treatments including mastectomy, nipple-sparing mastectomy [20,21] or breast-conserving therapy followed by adjuvant radiation in most cases. Finally, in cases of "upstaging" from DCIS to invasive ductal carcinoma [22], the false diagnosis could erroneously alter therapeutic options.

Screening mammography is variably recommended to start between 40 and 50 years of age, depending on recommending society, local practices, and patient factors [23]. However, there is not a clear upper age limit for screening in healthy women, since the incidence of breast cancer remains high into the 80s. Nevertheless, screening mammography may be less beneficial in women aged 75 and older and breast cancer screening is probably not associated with a reduction in mortality due to breast cancer in this age group [9]. Considering that potential benefits of screening mammography among women aged ≥75 years old remain unclear [11], screening could be individualized in this age group.

Finally, it is well known that obesity after menopause is strongly associated with an increased risk of breast cancer [6]. Considering that the above postmenopausal patient had a normal body mass index, among other characteristics and risk factors, this could be included in her "calculated risk" of screening decisions after the age of 80.

## 4. Conclusions

Abnormal findings on screening mammography which are confirmed in magnetic resonance imaging with subsequent positive for DCIS frozen section reports are not always confirmed in the final histologic examination. Furthermore, considering that breast cancer screening does not seem to be associated with a reduction in mortality due to breast cancer after the age of 75, such a screening may be individualized in this age group.

**Author Contributions:** G.I. and E.A. conceived the initial idea, designed the study and prepared the paper. S.Z. and A.V. supplied the material for the study. S.Z., A.V., A.B., A.M., E.O., P.T. and G.I. contributed to the methodology of the study. G.I. selected and compared data related to MRI. All authors equally contributed to the manuscript writing. All authors have read and agreed to the published version of the manuscript.

**Funding:** This research received no external funding.

**Acknowledgments:** Special thanks to Demetrius Iatrakis (National [Metsovian] Technical University of Athens) for technical assistance and text formatting.

**Conflicts of Interest:** The authors declare no conflict of interest.

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
