# Peer review of "“Harms” Associated with Breast Cancer Screening and Reliability of Frozen Section in Older Women: In the Case of an 80 Year Old Woman"

_reports, doi:10.3390/reports3020015_

Round 1

Reviewer 1 Report

Dear authrors.

This will be very interesting to readers.

It has something to be considered to get accepted.

  1. Conclusion is beyond the result. How did you connect this finding to the breast screening after the age of 75? (line 21-23, line 91-93)
  • So, make a new conclusion based on the result.

2. Reconcile the terms, please.

You usded BI_RADS_4C in line 37 but in line 49 , BI_RADS_IVC was used.

Maybe BI_RADS_4C is right. Check others.

3. Associated with No 1 critic, Line 78-84 is also beyond the this result.

Setting the upper age limit for cancer screening is the question in respect of mass or public, not personal.

If someone who is over 75 years old got the mammogram in personal and is suspected to have cancer, then what is next step? Of course, furher evaluations have to be followed to confirm.

4. In line 70, Frozen section -> FS

Author Response

  1. Conclusion is beyond the result. How did you connect this finding to the breast screening after the age of 75? (line 21-23, line 91-93)

So, make a new conclusion based on the result.

This is a conclusion derived from the whole manuscript because:

  1. the age of the woman is 80 years
  2. Continuing screening mammograms in older ages increases the possibility of false positive results (see line 33)
  3. Furthermore, this is a conclusion supported from related literature (Reference 11 etc.)

  1. Reconcile the terms, please.

You usded BI_RADS_4C in line 37 but in line 49 , BI_RADS_IVC was used.

Maybe BI_RADS_4C is right. Check others.

Changed (although both are used interchangeably)

  1. Associated with No 1 critic, Line 78-84 is also beyond the this result.

Setting the upper age limit for cancer screening is the question in respect of mass or public, not personal.

If someone who is over 75 years old got the mammogram in personal and is suspected to have cancer, then what is next step? Of course, furher evaluations have to be followed to confirm.

This is strictly related to the result, because EVERYTHING for this woman (further imaging, surgery, be informed about the initial positive histology could be avoided without the initial screening mammography (for which the value is debated).

  1. In line 70, Frozen section -> FS

As a matter of style, most journals advise authors not to start any sentence with an abbreviation...

Reviewer 2 Report

The manuscript reports a case of presumed DCIS that appeared positive on mammography and MRI but turned out to be a radial scar on pathology. The subject matter of the manuscript is quite mundane - cases like this occur often in typical clinical practice. The manuscript offers no support to guide controversy one way or another, nor is there a new perspective here. We expect that a fraction of cases will be false positives on imaging. We know that radial scars may look like DCIS. Women with presumed diagnoses of radial scars will undergo surgical excision realizing full well that the case may be negative for DCIS on pathology. There is nothing that should have been done differently here. No learning point. To add to this, the manuscript is not really citing the larger scale trials, nor is providing a clear consensus management algorithm.

These case reports can be useful where the cases are mundane such as the one submitted here, however such reports are only useful where they properly review presentation, imaging findings, clinical correlates, go into treatment algorithm, and present relevant pathology, epidemiology and controversies. I feel this just did not happen here to the extent that would make the manuscript valuable.

The manuscript suffers heavily from poor language use. The illustrations are very poor. The MRI images are not generally labelled in the figure legend for sequence type, presence or absence of contrast, additional findings in the right breast are not even mentioned.

If the authors can rectify these innumerable flaws, maybe the manuscript could be acceptable.

Below are just a few issues I noted in detail while I was reading the manuscript.
line 16: "cancerin" missing space
line 18: should read "with frozen section reports positive for DCIS" - it's unclear what you are actually stating.
line 21: instead of "verified", use "confirmed"
line 23: I don't understand how this sentence flows from the rest of the case. A signle anecdote will never provide adequate support for sweeping statements on screening. There is a lot of good literature on that. Either delete it or reference it or make it less categorical.
line 27: the diagnosis was not increased, the detection rate was increased, the incidence has increased. The diagnosis is the same - it's the name of the disease, not the number of cases.
lines 27-34: Please reword. What you are trying to say is worded extremely awkwardly.
line 37: need reference specifically to the statement that most DCIS is BIRADS 4C
line 44: 80 year old patient, not patient of 80 years old
line 69: there are several trials for preoperative MRI use, and you probably should look them up and cite them. That there is no consensus on preoperative MRI is correct.
line 69: "its integration", not "it's integration". Its means "belonging/referring to it". It's means "it is".
line 78: The lower age limit is actually not universally accepted. It is different in many areas of the world and varies between 40yo to 50yo. Please review the ACOG, ACR, NCCN, ACS, ASBS, ASCO, AAFP, ACP, CFTPHC, USPSTF, and the European societies before making such statements.

Author Response

The manuscript reports a case of presumed DCIS that appeared positive on mammography and MRI [and frozen section] but turned out to be a radial scar on pathology. The subject matter of the manuscript is quite mundane - cases like this occur often in typical clinical practice [The second and the last author are president and member respectively of the Greek-French oncology council, including members of the Georges Pompidou European Hospital, which is under the aegis of the AP-HP university hospital trust. Furthermore, the last author is a member of the oncology council of one of the largest gynaecologic clinics of Greece, REA. After hundreds of presented cases, this is the first case of initially “triple positive” which finally proved negative]. The manuscript offers no support to guide controversy [The meaning of “guide controversy” is unclear, as it is, including this occurrence, a hapax legomenon in all of medical literature] one way or another, nor is there a new perspective here [There are new perspectives here. For example, the perspective of stopping screening in older patients]. We expect that a fraction of cases will be false positives on imaging [Our case was positive on imaging and frozen section]. We know that radial scars may look like DCIS [already mentioned in the text]. Women with presumed diagnoses of radial scars will undergo surgical excision realizing full well that the case may be negative for DCIS on pathology [This is irrelevant, as the frozen section was positive]. There is nothing that should have been done differently here [This case is mentioned not because we did something differing from relevant guidelines, as doing so could be dangerous for the patient, but because the results of all previous exams, including frozen section, were pointing towards malignancy]. No learning point [At least two learning points: 1. positive imaging results supported by positive frozen section results could be misleading. 2. the upper age of screening mammography should be re-examined]. To add to this, the manuscript is not really citing the larger scale trials [Conclusions of “larger scale trials” are already summarized and included in cited references. Citing all such trials would distract from the focus of this report.], nor is providing a clear consensus management algorithm. [Even meta-analyses face difficulties to provide reliable algorithms. This is an interesting case report intending to emphasize the aforementioned learning points]. 

These case reports can be useful where the cases are mundane such as the one submitted here, however such reports are only useful where they properly review presentation, imaging findings, clinical correlates, go into treatment algorithm, and present relevant pathology, epidemiology and controversies [All of these points already answered and/or included in the study]. I feel this just did not happen here to the extent that would make the manuscript valuable [Having answered the above points, it can be concluded that the aforementioned “extent” is fully covered].

The manuscript suffers heavily from poor language use [Corrections have been made]. The illustrations are very poor [The illustrations are actually “better” than the original available data, scanned at 300 dpi and ready to print. Nevertheless, the quality of the illustrations does not affect the validity of this report. The imaging and frozen section reports were positive for malignancy]. The MRI images are not generally labelled in the figure legend for sequence type, presence or absence of contrast, additional findings in the right breast are not even mentioned [The most representative MRI slice was selected, and the lesion is clearly visible].

If the authors can rectify these innumerable review flaws [All flaws have been addressed; see above], maybe the manuscript could be acceptable.

Below are just a few issues I noted in detail while I was reading the manuscript.

line 16: "cancerin" missing space [Corrected].

line 18: should read "with frozen section reports positive for DCIS" - it's unclear what you are actually stating [The sentence has been reformulated].

line 21: instead of "verified", use "confirmed". [Corrected].

line 23: I don't understand how this sentence flows from the rest of the case. A signle anecdote will never provide adequate support for sweeping statements on screening. There is a lot of good literature on that. Either delete it or reference it or make it less categorical. [References are not included in the Abstract. See Reference 11: Elmore JG. Screening for breast cancer: Strategies and recommendations. UpToDate 2020 (“screening mammography may be less beneficial in women aged 75”]. 

line 27: the diagnosis was not increased, the detection rate was increased, the incidence has increased. The diagnosis is the same - it's the name of the disease, not the number of cases. [No major reason to disagree. However, this is the exact wording used in reference no. 5].

lines 27-34: Please reword. What you are trying to say is worded extremely awkwardly. [It’s perfectly clear. In every mammography, there is a possibility to take a false positive result. Increasing the number of mammographies, that possibility is increased].

line 37: need reference specifically to the statement that most DCIS is BIRADS 4C. [It’s written in the same reference (12)].

line 44: 80 year old patient, not patient of 80 years old [Corrected].

line 69: there are several trials for preoperative MRI use, and you probably should look them up and cite them. That there is no consensus on preoperative MRI is correct [Please read carefully the following text: Because MRI is so sensitive, it was assumed that preoperative MRI would estimate the extent of disease more accurately than conventional imaging, thereby improving surgical planning (eg, prompting a change to mastectomy when breast-conserving therapy had been previously considered [79]) and enabling surgeons to better obtain clean margins in breast-conserving surgery. However, available data have shown that routine preoperative breast MRI has not improved overall survival outcomes, improved the rate of breast conservation surgery achievement, or lowered locoregional recurrence rates [80-93]. UpToDate 2020].

line 69: "its integration", not "it's integration". Its means "belonging/referring to it". It's means "it is" [Corrected].

line 78: The lower age limit is actually not universally accepted. It is different in many areas of the world and varies between 40yo to 50yo. Please review the ACOG, ACR, NCCN, ACS, ASBS, ASCO, AAFP, ACP, CFTPHC, USPSTF, and the European societies before making such statements [The sentence has been reformulate

Round 2

Reviewer 1 Report

All issues that reviewer pointed out are soloved.

OK, it has enough qualitification for publication.

Author Response

Thank you very much for your comments

Reviewer 2 Report

I do not believe that the authors have sufficiently addressed the criticism and instead have opted to be quite defensive rather than following reason and logic. The authors invoke their membership and presidency of various societies, but this does not make their statements, leaps of logic, or embarrassing image quality any better.

1 I will reiterate - your 'hapax legomenon' of having DCIS diagnosis on frozen section pathology which was then overturned in permanent section histologic analysis (probably a few days later) is irrelevant to management and logically contradicts your conclusion that "screening may be less beneficial in this age group". False positive core biopsies and frozen sections do happen, in some conditions/tissues more than others. If having a false positive biopsy in general is a "hapax", and we know that frozen section analysis is not always perfect, among numerous studies, consider e.g. https://www.scielo.br/scielo.php?script=sci_arttext&pid=S1676-24442018000500319 then it's even a greater argument to excise these lesions for older women and trust original frozen section analysis - until final histology is there in a few days. Unfortunately, you make some kind of a logical leap connecting this case with "screening may be less beneficial in this age group" in your conclusions. This is illogical. Stopping screening could be right for some older women in the appropriate setting, but to the whole train of thought is simply illogical here. 

Authors should simply report that a frozen section was a false positive and your pathologists caught the false positive on final permanent histologic evaluation. Your conclusions about screening are completely irrelevant and completely derail an otherwise perfectly acceptable, if a little pedestrian, report.

2 Some efforts were made to make the manuscript somewhat more readable. Some sloppy and unreadable passages remain, however. Some things that caught my eyes:

Title: Instead of "80 years old woman" use "80 year old woman". I would not care normally, but it's just jarring to have this in your title.

line 17 "Actually, abnormal mammographic and magnetic resonance imaging findings with positive for DCIS frozen section were initially reported." should read something like "Mammographic and magnetic resonance imaging findings were concordant with the initial frozen section biopsy result of DCIS."

line 47 I don't know if the Greek-French oncology council adheres to the BIRADS lexicon, but they do use the BIRADS ratings without using the proper descriptors. Nearly all the descriptors seem to be a result of translation attempts, e.g. 'architectural disturbance' is actually 'architectural distortion' - see https://www.acr.org/-/media/ACR/Files/RADS/BI-RADS/BIRADS-Reference-Card.pdf . 

3 You can't simply wish image quality and image description issues away. You took a cell phone, didn't turn off flash, then took a photo of a laser printer-made MRI image to make Figure 2A. You likely did the same thing with Figure 2B and 2C, but this time you decided to turn off flash. This is simply sloppy. There is a "Print Screen" button. There is the ability to export DICOM images and view them in dedicated software and create publication-quality images. You should do this before resubmitting. If the images are irrelevant to the case, as you imply in your response, why are you including them?

4 Rewording of line 78 is misleading and can be taken by some to mean that screening mammography should be performed only after age 50. You should reword this to acknowledge variability in practices. This is a bitterly debated topic. Saying something acknowledging this would be advisable: "Screening mammography is variably recommended to start between 40 and 50 years of age, depending on recommending society, local practices, and patient factors".

5 I feel that the paragraph on line 85 doesn't add much. Age and gender are among the strongest risk factors. The fact that someone did not have one of 10 risks does not somehow negate the possibility of having breast cancer, especially when they have at least two of the strongest risk factors. If the authors have a counterpoint, they should add a reference.

6 Line 90. Use "confirmed" instead of "verified". Again, your conclusions are not only unsupported by data, but also are completely opposite to what your data argues, see above.

7 line 69: there are several trials for preoperative MRI use, and you probably should look them up and cite them. A good start would be to read your UpToDate text to get citations, then read some abstracts on PubMed, and then cite the relevant ones in your submission. 

Author Response

1 I will reiterate - your 'hapax legomenon' of having DCIS diagnosis on frozen section pathology which was then overturned in permanent section histologic analysis (probably a few days later) is irrelevant to management and logically contradicts your conclusion that "screening may be less beneficial in this age group". False positive core biopsies and frozen sections do happen, in some conditions/tissues more than others. If having a false positive biopsy in general is a "hapax", and we know that frozen section analysis is not always perfect, among numerous studies, consider e.g. https://www.scielo.br/scielo.php?script=sci_arttext&pid=S1676-24442018000500319 then it's (included in the references) even a greater argument to excise these lesions for older women and trust original frozen section analysis - until final histology is there in a few days. Unfortunately, you make some kind of a logical leap connecting this case with "screening may be less beneficial in this age group" in your conclusions. This is illogical. Stopping screening could be right for some older women in the appropriate setting, but to the whole train of thought is simply illogical here. 

Authors should simply report that a frozen section was a false positive and your pathologists caught the false positive on final permanent histologic evaluation. Your conclusions about screening are completely irrelevant and completely derail an otherwise perfectly acceptable, if a little pedestrian, report.

2 Some efforts were made to make the manuscript somewhat more readable. Some sloppy and unreadable passages remain, however. Some things that caught my eyes:

Title: Instead of "80 years old woman" use "80 year old woman". I would not care normally, but it's just jarring to have this in your title. Replaced.

line 17 "Actually, abnormal mammographic and magnetic resonance imaging findings with positive for DCIS frozen section were initially reported." should read something like "Mammographic and magnetic resonance imaging findings were concordant with the initial frozen section biopsy result of DCIS." Replaced.

line 47 I don't know if the Greek-French oncology council adheres to the BIRADS lexicon, but they do use the BIRADS ratings without using the proper descriptors. Nearly all the descriptors seem to be a result of translation attempts, e.g. 'architectural disturbance' is actually 'architectural distortion' - see https://www.acr.org/-/media/ACR/Files/RADS/BI-RADS/BIRADS-Reference-Card.pdf. Replaced.

3 You can't simply wish image quality and image description issues away. You took a cell phone, didn't turn off flash, then took a photo of a laser printer-made MRI image to make Figure 2A. You likely did the same thing with Figure 2B and 2C, but this time you decided to turn off flash. This is simply sloppy. There is a "Print Screen" button. There is the ability to export DICOM images and view them in dedicated software and create publication-quality images. You should do this before resubmitting. If the images are irrelevant to the case, as you imply in your response, why are you including them? Replaced.

4 Rewording of line 78 (new lines: 81-82) is misleading and can be taken by some to mean that screening mammography should be performed only after age 50. You should reword this to acknowledge variability in practices. This is a bitterly debated topic. Saying something acknowledging this would be advisable: "Screening mammography is variably recommended to start between 40 and 50 years of age, depending on recommending society, local practices, and patient factors". Rephrased.

5 I feel that the paragraph on line 85 (new line: 91) doesn't add much. Age and gender are among the strongest risk factors. The fact that someone did not have one of 10 risks does not somehow negate the possibility of having breast cancer, especially when they have at least two of the strongest risk factors. If the authors have a counterpoint, they should add a reference. Rephrased.

6 Line 90 (new line: 95). Use "confirmed" instead of "verified". Again, your conclusions are not only unsupported by data, but also are completely opposite to what your data argues, see above. Replaced. Rephrased (line: 98).

7 line 69: there are several trials for preoperative MRI use, and you probably should look them up and cite them. A good start would be to read your UpToDate text to get citations, then read some abstracts on PubMed, and then cite the relevant ones in your submission. Relevant references were cited (lines: 70-73).

Round 3

Reviewer 2 Report

There have been some good revisions, and while the paper is far from perfect, it is passable with one correction that must happen.

I am not sure if the authors have a medical imaging expert on board, but the conclusions made regarding benefits of preoperative MRI are incorrect. The authors are making some claims with rather old citations which have not been updated for the past 7-8 years.

As a result, the authors are making incorrect statements about preoperative breast MRI. The authors have to review and revise their statements about breast MRI. For an example of a recent citation that takes into consideration the current state of research on preoperative breast MRI, authors should see:

https://pubs.rsna.org/doi/10.1148/radiol.2020191535

and the annotation to this study at

https://pubs.rsna.org/doi/10.1148/radiol.2020200076

which nicely summarizes the controversy.

Even references 2-3 years subsequent to the ones cited by the authors demonstrate similar results regarding the benefits of preoperative MRI, but strangely, the authors did not even acknowledge the existence this research, e.g. https://www.ncbi.nlm.nih.gov/pmc/articles/PMC4900728/

which similarly demonstrates benefit of preoperative MRI, which drops the rates of repeat surgery in the patients receiving preoperative MRI three fold.

If they can correct their incorrect statements and include the appropriate citations with an appropriate statement, that should be sufficient for me. eg: "The use of preoperative MRI is controversial. While prior studies indicated limited benefit of preoperative MRI, emerging evidence demonstrates lower odds of positive resection margins and lower odds of repeat surgery in patient groups undergoing preoperative breast MRI."

Author Response

Your statement was included in the discussion («While prior studies indicated limited benefit of preoperative MRI, emerging evidence demonstrates lower odds of positive resection margins and lower odds of repeat surgery in patient groups undergoing preoperative breast MRI»).